# SwapPrompt: Test-Time Prompt Adaptation for Vision-Language Models

**Xiaosong Ma**
Department of Computing
The Hong Kong Polytechnic University
Hong Kong, China
`xiaosong16.ma@connect.polyu.hk`

**Jie Zhang** *
Department of Computing
The Hong Kong Polytechnic University
Hong Kong, China
`jie-comp.zhang@polyu.edu.hk`

**Song Guo**
Department of Computer Science and Engineering
Hong Kong University of Science and Technology
Hong Kong, China
`songguo@cse.ust.hk`

**Wenchao Xu**
Department of Computing
The Hong Kong Polytechnic University
Hong Kong, China
`wenchao.xu@polyu.edu.hk`

## Abstract

Test-time adaptation (TTA) is a special and practical setting in unsupervised domain adaptation, which allows a pre-trained model in a source domain to adapt to unlabeled test data in another target domain. To avoid the computation-intensive backbone fine-tuning process, the zero-shot generalization potentials of the emerging pre-trained vision-language models (e.g., CLIP, CoOp) are leveraged to only tune the run-time prompt for unseen test domains. However, existing solutions have yet to fully exploit the representation capabilities of pre-trained models as they only focus on the entropy-based optimization and the performance is far below the supervised prompt adaptation methods, e.g., CoOp. In this paper, we propose SwapPrompt, a novel framework that can effectively leverage the self-supervised contrastive learning to facilitate the test-time prompt adaptation. SwapPrompt employs a dual prompts paradigm, i.e., an online prompt and a target prompt that averaged from the online prompt to retain historical information. In addition, SwapPrompt applies a swapped prediction mechanism, which takes advantage of the representation capabilities of pre-trained models to enhance the online prompt via contrastive learning. Specifically, we use the online prompt together with an augmented view of the input image to predict the class assignment generated by the target prompt together with an alternative augmented view of the same image. The proposed SwapPrompt can be easily deployed on vision-language models without additional requirement, and experimental results show that it achieves state-of-the-art test-time adaptation performance on ImageNet and nine other datasets. It is also shown that SwapPrompt can even achieve comparable performance with supervised prompt adaptation methods.

## 1 Introduction

When there is a discrepancy between the distribution of the training data and the testing data, the generalization performance of deep neural networks can be compromised [1, 2, 3]. The focus of domain adaptation is to construct models that can adapt to variations in data distribution by

---

*Corresponding author

37th Conference on Neural Information Processing Systems (NeurIPS 2023).

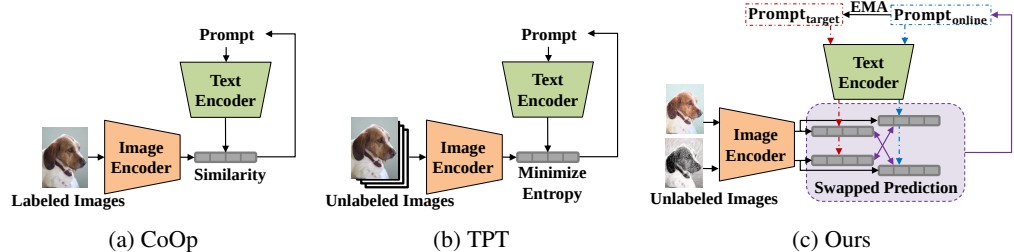

Figure 1: **Comparison on V-L model architectures.** (a) CoOp adapts prompt on labeled data. (b) TPT optimizes prompt by minimizing marginal entropy. (c) Ours SwapPrompt leverages self-supervised contrastive learning to facilitate test-time prompt adaptation.

transferring knowledge from a source domain to a new related target domain, which usually requires both the source and target domain data during the training phase [4, 5]. However, in practical scenarios, it is common to have only the model after it has been trained in the source domain, while without the access to the source data, or the authorization to alter the original training procedure [6, 7, 8]. To address this problem, Test-time adaptation (TTA) [9, 10] has been proposed and shown its potential to adapt models to target/unseen domains by only leveraging the unlabeled test data streams. Existing works have developed techniques such as entropy minimization [7, 11], class prototypes [12, 13], image generation [14, 15], and self-supervised training [10], which have already demonstrated superior performance.

Although traditional model-based TTA methods are shown effective, they typically rely on computationally intensive tuning to the parameters of the model backbone. The situation would be even worse with the advent of vision-language pre-trained models (e.g., CLIP [16], CoOp [17] and CoCoOp [18]), that have a massive number of parameters and are difficult to optimize. Therefore, it is promising to explore efficient techniques to fine-tune only a small set of parameters for adapting models to novel domains during testing while keeping the backbone fixed. Pre-trained vision-language models, which are trained on a significant amount of image-text pairs, have introduced a powerful paradigm that provides fresh insights for tackling this issue. A straightforward way is to utilize the strong zero-shot capabilities of the pre-trained vision-language models to discriminate various domains of test data via the fine-tuning over the labeled data of downstream tasks, however such way may not be feasible in TTA scenarios where the labeled downstream data is unavailable. Shu et al. [19] propose the test-time prompt tuning (TPT) to address the label scarcity problem in test-time. Nevertheless, it may lead to a risk of over-trust in the model (i.e., generating high confidence for a wrong result) from directly minimizing the entropy to tuning instance-specific prompts. The prediction confidence of TPT and proposed SwapPrompt can be found in appendix.

To this end, we propose SwapPrompt, a novel test-time prompt adaptation method as illustrated in Figure 1. Unlike previous approaches, SwapPrompt leverages a self-supervised contrastive learning strategy in the test domain, which consists of two key components: *exponential moving average (EMA) prompt* and *prompt swapped prediction mechanism*. The EMA mechanism employs a dual prompts paradigm: the target prompt and the online prompt. We optimize the online prompt while the target prompt is gradually updated through a slow-moving average process, which incorporates past information to increase stability and effectiveness. The prompt swapped prediction mechanism is inspired by the unsupervised learning method SwAV [20]. Based on an augmented view of image and the online prompt, SwapPrompt predicts the class assignment of an augmented view of the same image. This enables the online prompt to learn more representation knowledge. The rationale behind the swapped prediction strategy is that two different augmentations of the same image should have similar class assignments. The contrastive representation learning approach is leveraged to generate better decision boundaries [21].

In addition to the loss function of self-supervised contrastive learning, we employ the conventional cross-entropy loss as in CLIP and CoOp, which tune the prompt with high-confidence pseudo labels generated by the zero-shot CLIP. Our approach can also be employed for online test-time scenarios, where test data arrive in a flow of mini-batches. We break down the operation performed on all test data into multiple mini-batches, which is discussed in detail in the experimental section. We evaluate our method on various test-time adaptation benchmarks, including ImageNet and four natural

distribution shift datasets based on it, as well as nine fine-grained classification datasets. Experiment results show that our method achieves state-of-the-art test-time adaptation performance. We present our main contributions as follows:

- We propose SwapPrompt, a novel test-time prompt adaptation method that employs a self-supervised contrastive learning strategy, enabling prompts to better adapt to downstream image classification tasks.
- To the best of our knowledge, we are the first to apply unsupervised representation learning in prompt adaptation for pre-trained vision-language models. We introduce EMA prompt and prompt swapped prediction strategies, which enable the prompt to learn more knowledge from the powerful representation capabilities of pre-trained models.
- We conduct extensive experiments on ImageNet and its four variants, as well as nine other image classification datasets The empirical evaluation shows that our approach significantly outperforms current TPT methods and can even compete with supervised prompt adaptation methods on most datasets.

## 2   Related Work

**Test-Time Adaptation.** Test-time adaptation refers to reducing the performance gap when a source model is deployed on a different target domain of test data. The challenge of this issue is that only the source model and unlabeled test data are available, the training process and source data should not be accessed. Many solutions have been proposed to solve this problem, i.e., minimizing the entropy of the model's predictions [7, 11], maintaining a set of dynamically prototypes and measuring the similarity between test samples to each prototype [12, 13], generating new data similar to the target domain to assist model adaptation [14, 15] and utilizing the idea of self-supervised training to improve the generalization capability [10]. However, test-time adaptation in vision-language model is under-explored. Recently, Shu et al. [19] propose test-time prompt tuning (TPT) to extend the old entropy minimization method to vision-language model, but it is limited in practice due to the potential obvious over-confidence risk on predictions.

**Prompt Learning in Vision-Language Models.** Prompt learning first emerged in the field of natural language processing (NLP), aiming to enhance the performance of pre-trained models by utilizing different prompts. With the advent of vision-language models that integrate both visual and textual modalities, inspired by prompt learning in NLP, CoOp [16] is proposed for prompt learning, which transforms hand-crafted prompts into learnable continuous prompts and tunes them to adapt to downstream tasks. CoCoOp [17], an improvement upon CoOp, employs a meta-net and image features to generate individual prompts for each image. Additionally, there are also some other methods such as CLIP-adapter [22] and Tip-adapter [23] that do not modify the prompts but instead add additional classification layers after the backbone models. What they have in common is that all of them heavily rely on a set of labeled data, making them unsuitable for test-time settings. Another line of work focuses on enabling prompt learning in an unsupervised manner during training process, i.e., unsupervised prompt learning (UPL) [24]. However, it simply extends pseudo-labeling methods to vision-language models without fully leveraging the powerful representation capabilities of pre-trained models.

## 3   Methodology

In this section, we first introduce the preliminary and problem definition of test-time prompt adaptation, then elaborate the proposed SwapPrompt framework that leverages self-supervised contrastive learning to facilitate prompt adaptation, which is shown in Figure 2. Finally, we present the workflow in Section 3.3.

### 3.1   Preliminary and Problem Definition

We focus on test-time prompt adaptation for pre-trained vision-language models (e.g. CLIP), where the model is trained on the source domain, but the test data belongs to the target domain. In this scenario, zero-shot CLIP with a general prompt (e.g., "a photo of a [CLS]") has shown barely acceptable zero-shot generalization ability. However, these hand-crafted prompts cannot fully extract

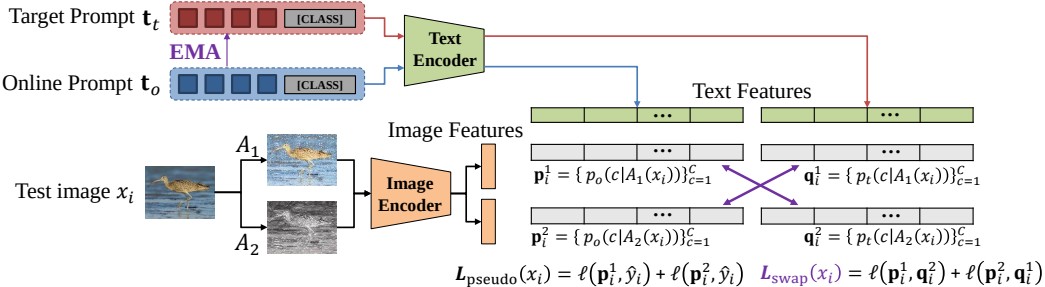

Figure 2: **Framework of the proposed SwapPrompt.** We use the text features generated by the target prompt as prototypes and assign the image feature of an augmented view of an image to these prototypes to obtain a soft class assignment. The online prompt is trained to predict this class assignment with a different augmented view of the same image. The EMA of online prompt is used to update the target prompt.

the rich knowledge learned by CLIP from large-scale and diverse pre-training datasets. Optimized prompts can further improve CLIP's ability to retrieve knowledge about the target domain. There have been some related works on supervised target domain data, including the well-known and effective CoOp method, part of our approach also incorporates the ideas of CoOp. Then we will briefly review CoOp as well as define the notation used in this paper.

**Context Optimization (CoOp).** CoOp is a prompt adaptation method based on CLIP. Similar to CLIP, CoOp includes an image encoder and a text encoder, which are denoted as $f(\cdot)$ and $g(\cdot)$, respectively. Let $\mathcal{D}_{\texttt{target}} = \{(x_i, y_i)\}_{i=1}^N$ be the dataset on the target domain, where $x_i$ is the $i$-th input data sample, $y_i$ is the corresponding label and $y_i \in \{1, 2, \ldots, C\}$ for a $C$-class classification problem, $N$ is the size of the dataset. Let $\mathbf{t}$ represent the learnable continuous prompt and $\{\mathbf{t}; c\}$ be the input of $c$-th class for the text encoder. We then define $\mathbf{z}_i = f(x_i)$ and $\mathbf{w}_c = g(\{\mathbf{t}; c\})$ as the output features embedding of image encoder and text encoder, respectively. The probability of $c$-th class for $x_i$ is computed as:

$$p(c|x_i) = \frac{\exp(\texttt{sim}(\mathbf{z}_i, \mathbf{w}_c)/\tau)}{\sum_{j=1}^C \exp(\texttt{sim}(\mathbf{z}_i, \mathbf{w}_j)/\tau)}, \tag{1}$$

where $\tau$ denotes the temperature parameter, $\texttt{sim}(\cdot)$ denotes the cosine similarity. For all training data, CoOp calculates the probabilities of all classes by Eq. 1 and minimizes the cross-entropy loss to tune the prompt.

**Test-Time Prompt Adaptation.** In the test-time scenario, labeled data from the target domain is not available, thereby the prompt cannot be optimized as in CoOp. Consider a test dataset $\mathcal{D}_{\texttt{test}} = \{x_i\}_{i=1}^N$ without label information. The objective of test-time prompt adaptation can be formulated as:

$$\mathbf{t}^* = \arg\min_{\mathbf{t}} \mathcal{L}(f, g, \mathbf{t}, \mathcal{D}_{\texttt{test}}), \tag{2}$$

where $\mathcal{L}$ is the cross-entropy loss function, our purpose is to design an unsupervised prompt adaptation method, facilitating prompt $\mathbf{t}$ compatibility with the target domain of $\mathcal{D}_{\texttt{test}}$, eliciting more target domain knowledge for CLIP.

## 3.2  Overview of SwapPrompt

In this section, we present our proposed test-time adaptation method SwapPrompt, which contains two key insights: *Exponential Moving Average (EMA) Prompt* and *Prompt Swapped Prediction*. As illustrated in Figure 2, SwapPrompt employs a dual prompts paradigm: the online and target prompts. These two types of prompts will be interacted and learnt from each other to adapt to the target domain by applying an EMA update strategy. Moreover, unlike most previous frameworks that only match one prompt to one image, SwapPrompt adopts a prompt swapped prediction mechanism to establish self-supervised representation learning in prompt adaptation with the goal of assigning similar classes for two different augmentations of the same image.

### 3.2.1 Exponential Moving Average Prompt

SwapPrompt's goal is to learn an online prompt $\mathbf{t}_o$ which can be used on the test dataset. As described previously, SwapPrompt has a target prompt $\mathbf{t}_t$ to guide the update of online prompt $\mathbf{t}_o$. The main motivation behind this design is: from a given target prompt $\mathbf{t}_t$, we can train a new and potentially improved prompt $\mathbf{t}_o$ through predicting the representation knowledge generated by the $\mathbf{t}_t$. By repeatedly using subsequent online prompts as new target prompts for further training, we can create a sequence of prompts that improves in quality over time. Practically, SwapPrompt uses a slowly moving exponential average of the online prompt as the target prompt, and we perform the following update after each training step:

$$\mathbf{t}_t = \epsilon \mathbf{t}_t + (1 - \epsilon)\mathbf{t}_o, \tag{3}$$

where $\epsilon \in [0, 1]$ is the decay rate of target prompt.

### 3.2.2 Prompt Swapped Prediction

In the field of self-supervised learning, cross-view prediction has been widely used in many existing works. These methods usually cast the prediction problem into a representation space, and then learn representations by predicting different augmented views generated from the same image. It is assumed that the different augmented views of an image should be relatively close in the representation space.

Among these methods, SwAV [20] proposes a different approach. Instead of directly enforcing consistent mapping between the image features in the representation space, SwAV clusters a set of image features, uses the cluster centroids as prototypes, and matches different augmentations of an image to these prototypes to compute its cluster assignment. By comparing their cluster assignments rather than their features, SwAV performs contrastive learning across multiple image views.

Inspired by SwAV, we assign an image's augmented view with prototypes to obtain its soft class assignment and predict this class assignment with another augmented view of the same image. This approach is well-suited for CLIP, as it has natural prototypes: text features outputted by the text encoder. Specifically, for all classes $\mathcal{Y} \in \{1, 2, \ldots, C\}$ in the test dataset $\mathcal{D}_{\mathtt{test}}$, target prompt $\mathbf{t}_t$ will form $C$ inputs $\{\mathbf{t}_t; c\}_{c=1}^{C}$ for text encoder, which will generate $C$ text features $\mathbf{w}_c^t = g(\{\mathbf{t}_t; c\})$ of different classes. Due to the supervised contrastive pre-training on large-scale image-text pairs, the text features generated by the CLIP have high similarity with the image features of the same class and low similarity with the text features of different classes. Therefore, these $C$ text features $\{\mathbf{w}_c^t\}_{c=1}^{C}$ can serve as a set of high-quality prototypes.

For an image $x_i$, we use two different data augmentation methods $A_1$ and $A_2$ to obtain two augmented views $A_1(x_i)$ and $A_2(x_i)$. The image features generated by the image encoder are $\mathbf{z}_i^1 = f(A_1(x_i))$ and $\mathbf{z}_i^2 = f(A_2(x_i))$, respectively. By applying Eq. 1 on prototypes $\{\mathbf{w}_c^t\}_{c=1}^{C}$, we can acquire the corresponding class assignments of two augmented image views:

$$\mathbf{q}_i^1 = \{p_t(c|A_1(x_i))\}_{c=1}^{C} \quad \text{and} \quad \mathbf{q}_i^2 = \{p_t(c|A_2(x_i))\}_{c=1}^{C}, \tag{4}$$

where the $p_t(c|A_1(x_i))$ and $p_t(c|A_2(x_i))$ can be expressed as:

$$p_t(c|A_1(x_i)) = \frac{\exp(\mathtt{sim}(\mathbf{z}_i^1, \mathbf{w}_c^t)/\tau)}{\sum_{j=1}^{C} \exp(\mathtt{sim}(\mathbf{z}_i^1, \mathbf{w}_j^t)/\tau)} \quad \text{and} \quad p_t(c|A_2(x_i)) = \frac{\exp(\mathtt{sim}(\mathbf{z}_i^2, \mathbf{w}_c^t)/\tau)}{\sum_{j=1}^{C} \exp(\mathtt{sim}(\mathbf{z}_i^2, \mathbf{w}_j^t)/\tau)}. \tag{5}$$

Similarly, the predictions generated by online prompt are defined as:

$$\mathbf{p}_i^1 = \{p_o(c|A_1(x_i))\}_{c=1}^{C} \quad \text{and} \quad \mathbf{p}_i^2 = \{p_o(c|A_2(x_i))\}_{c=1}^{C}. \tag{6}$$

We establish the prompt swapped prediction loss function for image $x_i$ as:

$$\boldsymbol{L}_{\mathtt{swap}}(x_i) = \ell(\mathbf{p}_i^1, \mathbf{q}_i^2) + \ell(\mathbf{p}_i^2, \mathbf{q}_i^1), \tag{7}$$

where $\ell$ is the function that measures the difference between the predictions and the class assignments. In this paper, we utilize the cross-entropy loss function. *i.e.,*

$$\ell(\mathbf{p}_i^1, \mathbf{q}_i^2) = -\sum_{c=1}^{C} p_t(c|A_2(x_i)) \log p_o(c|A_1(x_i)). \tag{8}$$

The online prompt $\mathbf{t}_o$ will be optimized by Eq. 7 while the target prompt $\mathbf{t}_t$ will not be updated by this loss function.

### 3.2.3 Prompt Optimization by Pseudo Label

In addition to utilizing self-supervised representation learning methods to optimize prompts, we also employ a set of labeled data, similar to CoOp, to optimize online prompts. However, unlike CoOp where the labels are available for target domain, the test data is unlabeled in the test-time scenario. Therefore, we first perform inference on the test data with hand-crafted prompts (e.g., "a photo of a [CLS]"), obtaining their pseudo-labels $\hat{\mathcal{Y}}_{\mathtt{test}} = \{\hat{y}_i\}_{i=1}^N$, and then employ Eq. 6 and cross-entropy loss as:

$$L_{\mathtt{pseudo}}(x_i) = \ell_{ce}(\mathbf{p}_i^1, \hat{y}_i) + \ell_{ce}(\mathbf{p}_i^2, \hat{y}_i). \tag{9}$$

It should be noted that the pseudo-labels obtained through this approach may contain noise. Therefore, we cannot apply Eq. 9 to all test data. The process of data selection is discussed in Section 3.3.

### 3.3 Algorithm Workflow

In this subsection, we illustrate the overall prompt adaptation process of SwapPrompt and the training method when facing online test samples.

Before training with Eq. 7, Eq. 9 and pseudo labels $\hat{\mathcal{Y}}_{\mathtt{test}}$, we need to perform data selection to filter out potential noisy pseudo labels. Specifically, we first employ the zero-shot CLIP and a hand-crafted prompt to obtain pseudo labels and classification confidences for the test data. Then, for each class, we only select the top $K$ test data with the highest confidence. These selected test data form the adaptation set $\mathcal{D}_{\mathtt{adapt}}$, which is a subset of $\mathcal{D}_{\mathtt{test}}$. For all $x_i \in \mathcal{D}_{\mathtt{adapt}}$, given trade-off hyper-parameters $\alpha$ and $\beta$, the following loss function will be used to do prompt adaptation:

$$L_{\mathtt{adapt}}(x_i) = \alpha L_{\mathtt{swap}}(x_i) + \beta L_{\mathtt{pseudo}}(x_i). \tag{10}$$

When target images arrive in a flow of mini-batches, i.e., the test data is online, we cannot sort the confidences of the entire test dataset. However, we can still perform confidence-based sorting on mini-batches to select the top $k$ ($k < K$) test data with the highest confidence, while keeping the rest of the training process unchanged. When new mini-batch test data arrives, the available test data are sorted by confidence again to obtain new $\mathcal{D}_{\mathtt{adapt}}$ for prompt adaptation.

## 4 Experiments

### 4.1 Experimental Setup

**Dataset.** We evaluate the proposed SwapPrompt over fourteen datasets, including ImageNet [25] and its four variants: ImageNet-V2 [26], ImageNet-A [27], ImageNet-R [28] and ImageNet-Sketch [29], and nine other publicly available image classification datasets used in CLIP: Caltech101 [30], DTD [31], Flowers102 [32], Oxford-Pets [33], UCF101 [34], StanfordCars [35], Food101 [36], EuroSAT [37] and SUN397 [38]. These datasets encompass a diverse range of visual classification tasks, including general objects, fine-grained categories, and even texture classification, forming a comprehensive benchmark. We only use the test data to do adaptation and also evaluate models with them.

**Baselines.** We compare the performance of SwapPrompt with the state-of-the-art methods. In addition to zero-shot CLIP [16], we also include TPT [19], a test-time prompt tuning method that minimizes the marginal entropy of test data; UPL [24], an unsupervised prompt learning approach and we make some modifications on it to suit the test-time setting; CoOp [17], a supervised few-shot prompt tuning method. We use some labeled data from the same domain as the test data during training this baseline, in order to use it as an upper bound performance of test-time prompt adaptation.

**Implementation Details.** In all experiments, we use the publicly available CLIP model with the ResNet-50 [39] visual encoder as the backbone model. Unless otherwise specified, the prompt is initialized randomly with 4 learnable tokens in SwapPrompt, UPL and CoOp. As for TPT, the prompt is initialized as the default one "a photo of a". When comparing the performance with baselines, we select the top 16 test data with the highest confidence to train SwapPrompt and UPL. For SwapPrompt, the decay rate of target prompt is 0.99, both $\alpha$ and $\beta$ are 1. We use the same image augmentation

Table 1: Comparison of test-time adaptation methods on 14 datasets. $\Delta$ denotes SwapPrompt's gain over the better one of UPL and TPT. '+ Online' denotes SwapPrompt with online test data.

| Method | Caltech101 | DTD | Flowers102 | Oxford-Pets | UCF101 | StanfordCars | Food101 | EuroSAT | SUN397 | ImageNet | ImageNet-V2 | ImageNet-A | ImageNet-R | ImageNet-Sketch |
|---|---|---|---|---|---|---|---|---|---|---|---|---|---|---|
| CoOp [17] | 88.76 | 54.62 | 83.98 | 87.44 | 66.71 | 61.83 | 73.79 | 61.68 | 64.33 | 61.23 | 55.29 | 23.41 | 56.96 | 35.64 |
| CLIP [16] | 85.13 | 42.16 | 65.40 | 83.05 | 61.15 | 55.65 | 74.23 | 37.60 | 58.55 | 58.18 | 51.36 | 21.69 | 55.98 | 33.33 |
| UPL [24] | 86.37 | 45.04 | 67.11 | 88.53 | 63.63 | 58.46 | 74.38 | 41.40 | 61.07 | 61.19 | 52.07 | 23.59 | 57.09 | 36.40 |
| TPT [19] | 87.22 | 42.17 | 65.42 | 84.60 | 61.18 | 58.49 | 74.88 | 43.82 | 61.46 | 60.74 | **54.35** | **26.24** | 58.72 | 35.02 |
| SwapPrompt | **89.90** | **47.34** | **70.22** | **89.14** | **65.66** | **59.60** | **75.08** | **46.64** | **63.93** | **61.80** | 53.94 | 24.46 | **60.88** | **38.21** |
| $\Delta$ | +2.68 | +2.30 | +3.11 | +0.61 | +2.03 | +1.11 | +0.20 | +2.82 | +2.47 | +0.61 | -0.41 | -1.78 | +2.16 | +1.81 |
| + Online | 89.69 | 46.40 | 68.12 | 88.97 | 64.52 | 58.88 | 75.66 | 42.45 | 63.36 | 61.41 | 52.93 | 24.42 | 60.25 | 38.13 |

method as SimCLR [40] to generate two different augmented images for an image. We optimize the prompts for 50 epochs with SGD optimizer and a cosine decay learning rate scheduler, the initial learning rate is 0.002. The batch size of images is 32 on all datasets.

We do all experiments on a workstation with an RTX 3090 GPU, a 3.5-GHZ Intel Core i9-11900K CPU and 64GB of RAM.

## 4.2 Performance Comparison

First, we compare our SwapPrompt with the baseline methods over fourteen benchmark datasets. The classification accuracy is listed in Table 1. It should be noted that CoOp is trained with labeled target domain data, and we use 4-shot data per category. From the results, it can be observed that our proposed SwapPrompt provides superior test-time adaptation performance than baselines on most datasets. We not only outperform the better baseline in UPL and TPT, but also very close even outperform CoOp on many datasets (e.g., Caltech101, Oxford-Pets, Food101, ImageNet, ImageNet-R and ImageNet-Sketch). Figure 3(a) shows that the average accuracy over 14 datasets for all baselines. SwapPrompt outperforms TPT and UPL by 2.31% and 2.17% accuracy, respectively.

**Strong performance in online test-time adaptation setting.** We also add results of SwapPrompt with online test data. Under this setting, we received mini-batches which only have a small part of data. For example, in DTD, a mini-batch has only 64 test data samples, less than 4% of the entire dataset. On most datasets, there is a slight decrease in accuracy because we cannot hold enough test data at the beginning of training to learn an appropriate prompt to classify the test data that come first. However, online SwapPrompt still outperforms UPL and TPT on most of datasets, as well as the average accuracy on all datasets in Figure 3(a). It should be noted that on Food101, the accuracy of online SwapPrompt is better. It is because the prompt in the intermediate stage is better than the one in the final stage. This could be attributed to the presence of high-confidence noise in pseudo labels during the final stage, which is discussed in section 4.3.

## 4.3 Ablation Study

In this subsection, detailed analyses are shown to help understand the superiority of our SwapPrompt method, including the trade-off between accuracy and efficiency, analysis on objective functions $L_{\text{swap}}$ and $L_{\text{pseudo}}$, the decay rate $\epsilon$ of target prompt, the effect of $K$ value in data selection, and the effect of prompt's context length and initialization.

**The Trade-Off between Accuracy and Efficiency.** The main factor which affects the efficiency of SwapPrompt is the prompt adaptation epoch. Figure 3(b) shows the relationship between the epoch and the average acuuracy of SwapPrompt on 5 datasets (Caltech101, DTD, Flowers102, Oxford-Pets and UCF101). It can be seen that SwapPrompt's accuracy increases quickly in the first 3 epoch, then

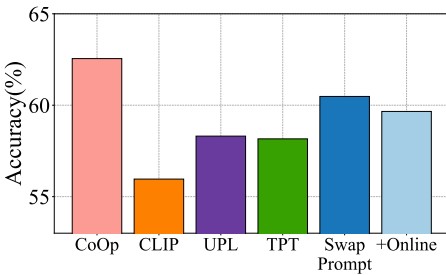
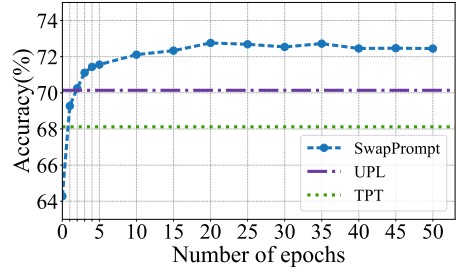

(a) Average accuracy on 14 datasets

(b) Accuracy of different epochs on 5 datasets

Figure 3: (a) The average accuracy on all 14 datasets, CoOp is compared as an upper bound. (b) The average accuracy of SwapPrompt on 5 datasets with different adpatation epochs, the accuracy of UPL and TPT is the final epoch average accuracy.

reaches its highest value around epoch 20 and stabilizes there. Thus, when the test time is limited, SwapPrompt can do a trade-off between the accuracy and adaptation epochs, e.g., only train 3 epochs for a quick inference. Noteworthy, SwapPrompt outperforms the final epoch accuracy of UPL at only epoch 2, and the accuracy of TPT at epoch 1.

**Analysis on Objective Functions.** We evaluate the two objective functions $L_{\text{swap}}$ and $L_{\text{pseudo}}$ of SwapPrompt on 5 datasets. Results in Table 2 gives a clear ablation study to demonstrate the effectiveness of our proposed objective functions. First, We use the UPL as the basic baseline, which has the confident test data selection and objective function $L_{\text{pseudo}}$. Then, UPL+AUG means that only adds image augmentation to the baseline, so that the $L_{\text{pseudo}}$ applies on 2 augmented image views. It can be observed that the accuracy has improved on all datasets, which demonstrates the benefits of data augmentation. Compare with SwapPrompt, UPL+Aug do not has the $L_{\text{swap}}$ function, and the performance is poor than SwapPrompt. Finally, the case of using all two objective functions, i.e., the complete SwapPrompt, has the best performance, which demonstrates that the loss function $L_{\text{swap}}$ for the swapped prediction mechanism can further improve prompt.

Table 2: Analysis of objective functions.

|  | ImageNet | Caltech101 | DTD | Flowers102 | Oxford-Pets | UCF101 | Average |
|---|---|---|---|---|---|---|---|
| UPL [24] | 61.19 | 86.37 | 45.04 | 67.11 | 88.53 | 63.63 | 68.65 |
| UPL+AUG | 61.30 | 87.75 | 46.04 | 68.43 | 87.67 | 65.15 | 69.39 |
| SwapPrompt | **61.80** | **89.90** | **47.34** | **70.22** | **89.14** | **65.66** | **70.68** |

It should be noted that we do not include the result of using only $L_{\text{swap}}$. Because considering the inherent generalization ability of CLIP, it is unfair to directly compare the performance of only $L_{\text{swap}}$ with $L_{\text{pseudo}}$. The standalone application of $L_{\text{swap}}$ does not sufficiently leverage the rich pre-trained knowledge embedded within CLIP. This can be seen as disregarding the pseudo-labels, which encapsulate the most pre-trained knowledge. Our proposed method, SwapPrompt, combines $L_{\text{swap}}$ and $L_{\text{pseudo}}$ resulting in improved performance compared to using $L_{\text{pseudo}}$ alone.

**Analysis on the Decay Rate of Target Prompt.** SwapPrompt updates the target prompt by a slow-moving average of online prompt, thus the target prompt represents a delay and more stable version of the online prompt, a higher decay rate indicates the retention of a greater amount of historical information. As is shown in Eq. 3, when the decay rate $\epsilon$ is 0, the target prompt is instantaneously updated to the online prompt at each step. It should be noted that using only one online prompt on $L_{\text{swap}}$ loss (i.e., $L_{\text{swap}}(x_i) = \ell(\mathbf{p}_i^1, \mathbf{q}_i^2) + \ell(\mathbf{p}_i^2, \mathbf{q}_i^1)$) cannot work because the gradient will collapse, the alternative is to maintain a target prompt that remains identical to the online prompt. (i.e., $\epsilon = 0$). When the decay rate $\epsilon$ is 1, the target prompt is never updated, and remains at a constant value corresponding to its initialization. In this case we initialize the target prompt as "a photo of a", thus it still has basic zero-shot generalization ability. There is a trade-off between updating the targets too often and updating them too slowly.

Table 3 shows the results of different decay rates on 6 datasets. When $\epsilon = 0$ and $\epsilon = 1$, the performance is poor than the other three eclectic values. All values of the decay rate between 0.9 and 0.999 yield have its best applicable datasets and 0.99 decay rate has a highest average accuracy. Besides that, compared with maintaining a fixed target prompt ($\epsilon = 1$), the EMA strategy can increase the average accuracy by 1.35%.

Table 3: Results for different decay rates $\epsilon$.

| $\epsilon$ value | ImageNet | Caltech101 | DTD | Flowers102 | Oxford-Pets | UCF101 | Average |
|---|---|---|---|---|---|---|---|
| 1 | 60.87 | 88.49 | 45.21 | 68.57 | 87.01 | 65.85 | 69.33 |
| 0.999 | **61.88** | 89.71 | 47.58 | 68.70 | 88.74 | **66.93** | 70.59 |
| 0.99 | 61.80 | **89.90** | 47.34 | 70.22 | **89.14** | 65.66 | **70.68** |
| 0.9 | 61.76 | 87.99 | **47.64** | **70.65** | 87.47 | 65.00 | 70.09 |
| 0 | 61.29 | 87.38 | 47.22 | 69.63 | 87.71 | 64.68 | 69.65 |

**Analysis on the swapped prediction mechanism.** Inspiring from the idea of self-supervised contrastive learning, SwapPrompt uses both image augmentation and EMA update strategies on target prompt. Both methods are effective and indispensable, and the combination of the two methods naturally leads to "swap prediction" (i.e., $\boldsymbol{L}_{\texttt{swap}}$).

There are two other loss functions which are similar to $\boldsymbol{L}_{\texttt{swap}}$: $\boldsymbol{L}_1(x_i) = \ell(\mathbf{p}_i^1, \mathbf{p}_i^2) + \ell(\mathbf{q}_i^2, \mathbf{q}_i^1)$ and $\boldsymbol{L}_2(x_i) = \ell(\mathbf{p}_i^1, \mathbf{q}_i^1) + \ell(\mathbf{p}_i^2, \mathbf{q}_i^2)$. However, compared with $\boldsymbol{L}_{\texttt{swap}}$, both $\boldsymbol{L}_1$ and $\boldsymbol{L}_2$ lost one of advantage. For $\boldsymbol{L}_1$, the two prompts will adapt independently without any interaction so the online prompt cannot be updated under the guidance of target prompt. The advantage of historical information will be lost. For $\boldsymbol{L}_2$, the contrastive phase is removed due to the lack of the approaching process between two augmented images. The advantage of image augmentation will diminish. Table 4 provides the performance of using those three loss functions on 5 datasets. The experimental setting is the same as Table 1 except for the part of $\boldsymbol{L}_{\texttt{swap}}$. The results indicate that $\boldsymbol{L}_{\texttt{swap}}$ outperforms $\boldsymbol{L}_1$ and $\boldsymbol{L}_2$.

Table 4: Results for using different loss function.

| | Caltech101 | DTD | Flowers102 | Oxford-Pets | UCF101 | Average |
|---|---|---|---|---|---|---|
| $\boldsymbol{L}_1$ | 87.38 | 47.22 | 69.63 | 87.71 | 64.68 | 71.32 |
| $\boldsymbol{L}_2$ | 88.45 | 46.69 | 69.28 | 87.30 | 64.46 | 71.24 |
| $\boldsymbol{L}_{\texttt{swap}}$ | **89.90** | **47.34** | **70.22** | **89.14** | **65.66** | **72.45** |

**Analysis on the Top-$K$ Confident Data Selection.** We have a data selection before the prompt adaptation to filter out potential noise pseudo labels. Only top $K$ confident test data will be used in test-time adaptation. Too much data not only will decrease model performance, but also slightly increased training time. Table 5 provides the performance of SwapPrompt with different $K$ values on 5 datasets, the result without data selection is also included. In general, larger values of $K$ show better performance, the highest accuracy across all 5 datasets is achieved when $K$ is set to 8 or 16, with $K = 16$ having the best average accuracy. On the other hand, the performance using the entire test data does not surpass the performance when data selection is employed. Data selection can increase the average accuracy by 2.5%. This indicates that the negative impact of noisy pseudo labels outweighs the positive effects.

**Analysis on the Context Length and Initialization of Prompt.** To explore whether our Swap-Prompt works equally well on prompts in different context lengths and initialization, we repeat experiments on 5 datasets by varying the context length from 4 to 8 to 16, and initializing with three different type.

The results of different context lengths are shown in Table 6(a), which indicates that having more context tokens sometimes leads to slightly decrease on accuracy. This is probably due to shorter prompt learned less overfitting from selected samples. SwapPrompt only uses a part of test data to do adaptation, too much parameters in prompts may cause overfitting on those data. Nevertheless,

Table 5: Results for different $K$ in data selection. 'None' denotes no data selection.

| $K$ value | Caltech101 | DTD | Flowers102 | Oxford-Pets | UCF101 | Average |
|---|---|---|---|---|---|---|
| 1 | 87.51 | 41.13 | 65.04 | 87.83 | 62.20 | 68.74 |
| 2 | 88.72 | 43.50 | 64.66 | 87.42 | 62.97 | 69.45 |
| 4 | 89.49 | 44.15 | 66.63 | 88.98 | 62.76 | 70.40 |
| 8 | **90.14** | **47.70** | 66.30 | 88.43 | 63.96 | 71.31 |
| 16 | 89.90 | 47.34 | **70.22** | **89.14** | **65.66** | **72.45** |
| None | 88.03 | 43.97 | 68.41 | 86.68 | 62.64 | 69.95 |

SwapPrompt still maintains advanced performance on different context lengths. In Table 6(b), the three different initialization are hand-craft: "a photo of a [CLS]", Pre-ImageNet: the prompt which is trained on 16-shot ImageNet with CoOp, and random initialization. We find that different initialization only slightly affects the final accuracy. Because of the effective adaptation on prompt, SwapPrompt demonstrates robust performance, which indicates that our method does not rely on any prior source domain prompt.

Table 6: Analysis on the context length and initialization of prompt.

(a) Results for different context lengths.

| Length | Caltech101 | DTD | Flowers102 | Oxford-Pets | UCF101 | Average |
|---|---|---|---|---|---|---|
| 4 | **89.90** | 47.34 | 70.22 | **89.14** | 65.66 | **72.45** |
| 8 | 88.24 | **47.46** | **70.77** | 88.79 | 65.62 | 72.18 |
| 16 | 87.71 | 46.39 | 70.12 | 88.43 | **66.48** | 71.83 |

(b) Results for different initialization.

| Initialization | Caltech101 | DTD | Flowers102 | Oxford-Pets | UCF101 | Average |
|---|---|---|---|---|---|---|
| Hand-craft | 89.53 | 46.92 | 70.32 | 88.92 | **65.98** | 72.33 |
| Pre-ImagNet | 89.65 | 47.15 | **70.44** | 89.03 | 65.34 | 72.36 |
| Random | **89.90** | **47.34** | 70.22 | **89.14** | 65.66 | **72.45** |

**Analysis on the sensitivity of hyper-parameters $\alpha$ and $\beta$.**   To explore the sensitivity of Swap-Prompt about hyper-parameters $\alpha$ and $\beta$, we conduct experiments with different values of $\alpha$ and $\beta$ on 5 datasets, other experimental settings are the same as Table 1. The results are shown in Table 7. It can be seen that SwapPrompt is not sensitive to the choice of hyperparameters $\alpha$ and $\beta$ in most cases. Results of SwapPrompt with different hyper-parameters settings in Table 7 still outperforms baselines in Table 1.

Table 7: Analysis on the sensitivity of hyper-parameters $\alpha$ and $\beta$.

| | Caltech101 | DTD | Flowers102 | Oxford-Pets | UCF101 | Average |
|---|---|---|---|---|---|---|
| $\alpha = 0.6, \beta = 1.4$ | 88.56 | 46.80 | 70.04 | 88.24 | 65.21 | 71.77 |
| $\alpha = 1.0, \beta = 1.0$ | **89.90** | **47.34** | 70.22 | **89.14** | 65.66 | **72.45** |
| $\alpha = 1.4, \beta = 0.6$ | 89.45 | 47.10 | **70.32** | 87.97 | **65.77** | 72.12 |

## 5   Conclusion

In this paper, we have investigated a novel test-time prompt adaptation method, SwapPrompt, to learn the prompt adapted to the test domain for pre-trained vision-language models. Specifically, we maintain an online prompt and an EMA updated target prompt which interact and learn from each other. A swapped prediction mechanism is designed to train the online prompt, enabling it to predict the target prompt's class assignment of the same image under a different augmented view. Without any other requirement, SwapPrompt can be easily deployed on the test-time of vision-language models. Extensive empirical experiments have been conducted over various datasets to verify the effectiveness and superior performance of SwapPrompt.

## Acknowledgements

This research was supported by fundings from the Key-Area Research and Development Program of Guangdong Province (No. 2021B0101400003), Hong Kong RGC Research Impact Fund (No. R5060-19, No. R5034-18), Areas of Excellence Scheme (AoE/E-601/22-R), General Research Fund (No. 152203/20E, 152244/21E, 152169/22E, 152228/23E), Shenzhen Science and Technology Innovation Commission (JCYJ20200109142008673).

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
