# OpenReview forum: "SwapPrompt: Test-Time Prompt Adaptation for Vision-Language Models"
_NeurIPS.cc/2023/Conference — NeurIPS 2023 poster_

### Official Review · Reviewer_3xii · 2023-06-27

**Soundness:** 3 good
**Presentation:** 2 fair
**Contribution:** 2 fair
**Rating:** 5
**Confidence:** 5

**Summary:**

Authors propose SwapPrompt for the CLIP-based test-time adaptation task. SwapPrompt consists of two loss functions, swapped prediction loss and pseudo label loss, with a prompt EMA strategy. Experimental results on 14 datasets validate that SwapPrompt outperforms several baselines.

**Strengths:**

1. The experimental results of the article look convincing.
2. Authors conduct a detailed experimental analysis of the proposed method.
3. The paper is well-organized.

**Weaknesses:**

1. The proposed method is too simple, authors add the prompt swapped prediction loss to the baseline UPL, which lack novelty in NeurIPS conference.
2. "A straightforward way is to .... tunning instance-specific prompts. " in Introduction is confusing. Zero-shot inference by CLIP does not need fine-tuning and TPT is not designed to provide a correct confindence. Authors' comments of the previous work are strange.
3. Eq.(3) is a line of code rather than a mathematical formula.
4. Authors should provide sensitivity analysis about hyperparameters α and β.
5. Both Eq.(8) and Eq.(9) use l to represent loss but they have different meaning, try to use different subscripts.
6. Ablation study should provide the analysis about only Lswap in Table 2.

**Questions:**

1. Could authors provide the results of FGVCAircraft dataset, which is used in CoOp and UPL experiments.
2. Which augmentation is used in the prompt swapped prediction loss? Which augmentation is used in calculation of the pseduo label? Are choises of the augmentation affect the swapped prediction loss?

**Limitations:**

No potential negative societal impact.

---

> ### Author Rebuttal · Authors · 2023-08-09
>
> Thanks for your valuable feedback and suggestions. We have tried to address all your concerns as follows:
>
> **W1:** "The proposed method is ... in NeurIPS conference."
>
> **A1:** Sorry for the insufficient explanation on the novelty of our proposed method. We would like to re-emphasize the novelty and contribution below:
>
> Firstly, we are the first to introduce the concept of unsupervised representation learning into test-time prompt adaptation in pre-trained vision-language models, which is achieved by designing a prompt swapped prediction loss.
>
> Additionally, the prompt swapped prediction loss proposes two mechanisms that are employed for the first time in this problem: EMA prompt and prompt swapped prediction strategies. Their combined application to address this specific issue is innovative.
>
> Furthermore, we extensively conduct experiments on ImageNet and its four variants, along with nine other image classification datasets. The empirical evaluation demonstrates that our approach significantly outperforms current TPT methods and even competes with supervised prompt adaptation methods on the majority of datasets.
>
> The simplicity of the SwapPrompt approach also facilitates its implementation and usage during test-time with vision-language models.
>
> **W2:** "A straightforward way is ... are strange."
>
> **A2:** We apologize for the lack of clarity in our expression. The original meaning of this sentence is to clarify that techniques such as CoOp and CoCoOp are not suitable for deployment during test time. Regarding TPT, it is true that its primary design does not cater to the provision of precise confidence estimates. Nevertheless, this limitation may hold significance in specific scenarios. The prediction confidence of TPT and SwapPrompt can be found in Table 2 of **Supplementary Material**.
>
> To improve the readability of the whole paper, we have carefully revised the presentation in the new version.
>
> **W3:** Eq.(3) is a line of code rather than a mathematical formula.
>
> **A3:** Thanks for the review but there seems to be a misunderstanding about the EMA strategy. We want to point out that Eq. (3) is exactly a mathematical formula rather than a line of code. The same equation can be found in many related literature, e.g., [1][2][3]. In the upcoming version, we will introduce subscripts to differentiate between the target prompts before and after the update, aiming to enhance the clarity of the expression of this formula.
>
> [1] Grill J B, Strub F, Altché F, et al. Bootstrap your own latent-a new approach to self-supervised learning[J]. NeurIPS, 2020, 33: 21271-21284.
>
> [2] Cai Z, Ravichandran A, Maji S, et al. Exponential moving average normalization for self-supervised and semi-supervised learning[C]//CVPR. 2021.
>
> [3] He K, Fan H, Wu Y, et al. Momentum contrast for unsupervised visual representation learning[C]//CVPR. 2020: 9729-9738.
>
> **W4:** The sensitivity analysis about hyperparameters α and β.
>
> **A4:** Thanks for your valuable comments. The following table presents the results.
> It can be seen that SwapPrompt is not sensitive to the choice of hyperparameters α and β in most case. It still outperforms baselines.
>
> |    | Caltech101  | DTD | Flowers102 | Oxford-Pets | UCF101 |
> |  :-  | :-:  | :-:  | :-:  | :-:  | :-:  |
> | α=0.6, β=1.4  | 88.56 | 46.80 | 70.04 | 88.24 | 65.21 |
> | α=1.0, β=1.0  | 89.90 | 47.34 | 70.22 | 89.14 | 65.66 |
> | α=1.4, β=0.6 | 89.45 | 47.10 | 70.32 | 87.97 | 65.77 |
>
> **W5:** "Both Eq.(8) and ... different subscripts."
>
> **A5:** Thanks for your valuable comments. We will change the $l$ in Eq.(9) to $l_{ce}$.
>
> **W6:** Ablation study should provide the analysis about only Lswap in Table 2.
>
> **A6:**  Thanks for your valuable comments. We have added ablation study to compare the performance of only using $L_{swap}$ with other versions. The experimental results can be seen in the following table.
>
> |  | Caltech101  | DTD | Flowers102 | Oxford-Pets | UCF101 |
> |  :-  | :-:  | :-:  | :-:  | :-:  | :-:  |
> | UPL  | 86.37 | 45.04 | 67.11 | 88.53 | 63.63 |
> | only $L_{swap}$  | 86.25 | 43.38 | 65.01 | 84.33 | 61.69 |
> | UPL+AUG | 87.75 | 46.04 | 68.43 | 87.67 | 65.15 |
> | SwapPrompt | 89.90 | 47.34 | 70.22 | 89.14 | 65.66 |
>
> Although the performance of only using Lswap is worse than UPL, it is in fact acceptable. Considering the inherent generalization ability of CLIP, it is unfair to directly compare the performance of only using $L_{swap}$ with $L_{pseudo}$.
>
> The standalone application of $L_{swap}$ does not sufficiently leverage the rich pre-trained knowledge embedded within CLIP. This can be seen as disregarding the pseudo-labels, which encapsulate the most pre-trained knowledge. Our proposed method, SwapPrompt, combines $L_{swap}$ and $L_{pseudo}$, resulting in improved performance compared to using $L_{pseudo}$ alone.
>
> **Q1:** The results of FGVCAircraft dataset.
>
> **A1:**  The following table presents the results of FGVCAircraft dataset. It can be seen that SwapPrompt still outperforms baselines.
> |    | CLIP | UPL | TPT | SwapPrompt|
> |  :-  | :-:  | :-:  | :-:  | :-:  |
> | FGVCAircraft  | 17.07 | 17.45 | 17.69 | 18.03 |
>
> **Q2:**  Augmentations used in SwapPrompt.
>
> **A2:** We use the same augmentation method as SimCLR in swapped prediction loss and the regular image pre-processing methods in calculation of the pseduo label, which is the same as CoOp. We added experiments using the same augmentation method as FixMatch[4] in swapped prediction loss, and the results demonstrate that the choices of  augmentation will slightly affect the swapped prediction loss.
>
> |  Augmentation | Caltech101  | DTD | Flowers102 | Oxford-Pets | UCF101 |
> |  :-  | :-:  | :-:  | :-:  | :-:  | :-:  |
> | SimCLR  | 89.90 | 47.34 | 70.22 | 89.14 | 65.66 |
> | FixMatch | 89.61 | 46.51 | 70.24 | 88.89 | 66.26 |
>
> [4] Sohn K, Berthelot D, Carlini N, et al. Fixmatch: Simplifying semi-supervised learning with consistency and confidence[J]. Advances in neural information processing systems, 2020, 33: 596-608.

---

> ### Author Response · Authors · 2023-08-14
> **Look Forward to Post-rebuttal**
>
> Dear Reviewer,
>
> Thank you very much for spending time providing constructive and valuable comments on our paper. We have carefully replied to your questions and presented the supplementary experiments in the response.
>
> We provided detailed studies to inspect the performance of SwapPrompt on different hyperparameters **α** and **β**, only $L_{swap}$ in Table 2, **FGVCAircraft** dataset, and different **augmentation methods**. The experimental results verify that SwapPrompt is a simple but stable and effective method. Besides, we also discussed the novelty and contribution of SwapPrompt.
>
> We would like to know if there are any additional clarifications or experiments that we can offer. We are looking forward to your feedback and sincerely invite you to update the score if concerns are addressed.
>
> Thanks a lot for your time again!
>
> Best, Authors

---

> ### Comment · Reviewer_3xii · 2023-08-20
>
> Thanks for the detailed answers to my questions. The author's reply solved most of my concerns.
> I hope the author can add corresponding experiments to the paper, especially the ablation study.
> Based on this, I decided to raise my rating to Borderline accept.

---

> > ### Author Response · Authors · 2023-08-21
> >
> > Dear reviewer,
> >
> > Thanks for your valuable comments on our paper. We will conduct a more comprehensive ablation study in the upcoming version.
> >
> > Thanks a lot for your time again!
> >
> > Best, Authors

---

### Official Review · Reviewer_V8X4 · 2023-07-01

**Soundness:** 4 excellent
**Presentation:** 3 good
**Contribution:** 4 excellent
**Rating:** 8
**Confidence:** 4

**Summary:**

This work presents a framework to perform test-time prompt adaptation for pre-trained vision-language models. In test-time settings, one has no access to labeled test data. Therefore, the authors employ self-supervised contrastive learning techniques on the test domain. In particular, this works proposes an exponential moving average prompt mechanism, and a prompt swapped prediction mechanism. The proposed approach attains state-of-the-art results on image classification benchmarks against other unsupervised test-time adaptation methods. It shows competitive results when compared with approaches that actually use labeled data during test-time (CoOp).

**Strengths:**

Originality

- This work presents a novel approach to tackle the following problem: vision-language models pre-trained on a source domain cannot reach their full potential when deployed on a new target domain where no labels are available. I find this a very relevant problem to tackle, with major repercussions on a wide variety of down-stream tasks.
- Its two proposed mechanisms for test-time adaptation leverage smart unsupervised learning techniques to overcome the lack of target domain labels. Although these mechanisms are not novel per se, their combination tacke this specific problem is.

Quality:

- The paper is well-written in its most part. See “Weaknesses” below.

Clarity:

- The proposed method is clear, and the authors do a good job in comparing their approach against previous works and problem statements.

Significance:

- Tackling the problem of test-time adapting prompts to leverage pretrained vision-language backbones when no labels are available is quite a major and interesting challenge.

**Weaknesses:**

Please have a native english speaker review the grammar.

- Sentences like “SwapPrompt’s goal is to learn an online prompt t_0 which can be used for test dataset.” (line 153) do not sound grammatically correct. You’re probably missing a “the” before dataset. And instead of using “for”, you should probably use “on”.
- In line 229, the sentence is very long and hard to read. The comma in this line (“prompt tuning method, we use some labeled data”) should not be a comma, but rather a full stop.
- Line 258 (”It should be noted that on Food101, the accuracy
258 of online SwapPrompt is better, it is because the prompt …”). Again, the sentence is too long. I would say you need a full stop after “better”.
- Line 315-317 (”This is probably due .. data”). This sentence is very poorly written. Please correct grammar. What do you mean by ”...shorter prompt learned less overfitting”? Also, it should be ”uses”, not “use”. In general, this sentence is again too long.
- Lines 320-323. Another too long sentence. Please rewrite into 2 sentences. Will make what you’re trying to say so much more clear.

**Questions:**

- Potential Ablation Study: one might argue that gathering a few groundtruth labels for the target domain is always a possibility. I understand this is not the scope of the paper. But I’d be curious to see how much this approach could improve if one has, say, 5-10 labeled images in the mini batches. Maybe one could downweight the auto-labeled samples in upweight the actual groundtruth samples? Just a potential ablation that could be interesting. Authors, please don’t tackle this ablation if you consider this not to be relevant and, especially don’t tackle this if it would take a long time to prepare. Some comments on this would be enough to satisfy my interest here. I think I’m just curious to see how much labeled data your approach would need to beat CoOp (ideally less, right?)
- Could you talk about how your approach could be integrated into downstream tasks, such as object detection? For instance, how could one integrate SwaptPromt into OWL-ViT (Vision Transformer for Open-World Localization)?
- Will the code become available? Could be interesting for reproducibility.

**Limitations:**

Limitations have been addressed implicitly. It might be a good idea to add a very small section that explicitly points out all the limitations that authors have identified. This is typically a helpful section for future researchers to get inspiration from for future work.

---

> ### Author Rebuttal · Authors · 2023-08-09
>
> Thank you so much for recognizing the value of our work and for the constructive comments, which have helped us a lot to improve the paper's quality and clarity.
>
> **W1-5:** Please have a native english speaker review the grammar.
>
> **A1-5:** We appreciate your highlighting of the grammatical errors and some unclear expression in our paper. We will rectify these issues in the new version and reduce the use of lengthy and complex sentences.
>
> **Q1:** Potential Ablation Study: one might argue that gathering a few groundtruth labels for the target domain is always a possibility. I understand this is not the scope of the paper. But I’d be curious to see how much this approach could improve if one has, say, 5-10 labeled images in the mini batches. Maybe one could downweight the auto-labeled samples in upweight the actual groundtruth samples? Just a potential ablation that could be interesting. Authors, please don’t tackle this ablation if you consider this not to be relevant and, especially don’t tackle this if it would take a long time to prepare. Some comments on this would be enough to satisfy my interest here. I think I’m just curious to see how much labeled data your approach would need to beat CoOp (ideally less, right?)
>
> **A1:** Thank you for presenting such a valuable experimental scenario.  It is true that labeled data may be scarce in many real-world applications due to the tremendous overhead and the requirement of corresponding expertise in data annotation. On this basis, some semi-supervised learning methods can be introduced to complement the current prompt-tuning framework. However, due to the time constraints for rebuttal, we cannot conduct detailed experiments on this matter. I believe our approach can achieve comparable performance to CoOp's requirements with less labeled data. We can incorporate some semi-supervised learning techniques to refine our solution. For example, we can fine-tune the prompt using labeled data and then use this refined prompt to generate pseudo-labels for unlabeled data. Alternatively, as you mentioned, giving more weight to ground truth samples is also a viable option.
>
> This is an intriguing question, and we appreciate your suggestions. We will consider this question in future research.
>
> **Q2:** Could you talk about how your approach could be integrated into downstream tasks, such as object detection? For instance, how could one integrate SwaptPrompt into OWL-ViT (Vision Transformer for Open-World Localization)?
>
> **A2:** We believe our approach can be integrated into existing state-of-the-art methods, which utilize vision-language models for downstream tasks. For instance, as you mentioned, OWL-ViT. As far as we know, OWL-ViT, similar to CLIP, undergoes training using image-text pairs, but it doesn't adapt the prompt during inference. This is where our SwapPrompt can be embedded, allowing fine-tuning of the prompt, followed by obtaining the final detection results.
>
> **Q3:** Will the code become available? Could be interesting for reproducibility.
>
> **A3:** Yes, we will open the code once the paper is published.

---

> > ### Comment · Reviewer_V8X4 · 2023-08-12
> >
> > Thank you.
> >
> > Yes, I think giving more weight to ground truth samples could be an interesting exploration. Consider introducing this in the final paper if accepted.
> >
> > Thanks for the clarifications.

---

> > > ### Author Response · Authors · 2023-08-14
> > >
> > > Dear reviewer,
> > >
> > > Thanks for your valuable comments on our paper. We will try our best to explore this question in future research.
> > >
> > > Thanks a lot for your time again!
> > >
> > > Best, Authors

---

### Official Review · Reviewer_N4YN · 2023-07-06

**Soundness:** 3 good
**Presentation:** 3 good
**Contribution:** 3 good
**Rating:** 5
**Confidence:** 4

**Summary:**

This paper studies an interesting and practical task “test-time adaptation for large vision and language models”. The authors propose a novel SwapPrompt method by introducing self-supervised contrastive learning into prompt tuning. The proposed dual prompts paradigm and swapped prediction mechanism enhance performance, achieving state-of-the-art results on ImageNet and nine other datasets. However, I still have several questions that need to be further clarified, as outlined below.

**Strengths:**

1. The proposed SwapPrompt method is novel in the TTA community. It successfully introduces self-supervised contrastive learning into Prompt Tuning-based methods.
2. Experiments are thorough and demonstrate the effectiveness of the proposed method.
2. The paper is well-written and easy to follow.

**Weaknesses:**

1. Two important ablation experiments are missing. $L_{swap}$ actually has two variants: $L_1 = l(p1, p2) + l(q1, q2)$ and $L_2 = l(p1, q1) + l(p2, q2)$. For these two variants, I think they are essentially the same as the consistency maximization used in TPT and MEMO. In this sense, comparison with $L_1$ and $L_2$ are necessary to verify the effectiveness of “Swap”.

2. More in-depth analyses/motivation regarding why “Swap Predictions” is working are preferred. The current explanations are not thorough enough. So I would like to see more details in the authors’ response.


**Questions:**

1. Could the authors summarize the advantages of Prompt Tuning over Image Encoder Tuning?

2. How about the performance of SwapPrompt if SwapPrompt also updates the parameters (e.g., affine parameters in norm layers) in Image Encoder?

3. Comparisons with “only one online prompt” + “consistency maximization (i.e., $l(p1, p2)$)” are also preferred.

4. For Table 4, could the authors also include the results of CLIP? Additionally, when the batch size is very small during online adaptation, e.g., 1, how to select data? And in this case, will the performance gain of SwapPrompt degrade a lot?

5. I am curious about the performance of SwapPrompt on corruption data (ImageNet-C)?

6. Could the authors report more efficiency comparisons with TPT? Such as GPU runtime and Memory. I think SwapPrompt would be much more efficient and this is also an advantage.

7. Will the authors release the source code after the acceptance of this work?

**Limitations:**

no limitations discussed.

---

> ### Author Rebuttal · Authors · 2023-08-09
>
> We would like to thank the reviewer for the constructive and detailed comments, we have tried to address all your concerns one by one.
>
> **W1&W2:**  "Two important ... in the authors’ response."
>
> **A1&A2:**  Thanks for your valuable comments. We will first explain why "swap prediction" works: inspiring from the idea of self-supervised contrastive learning, our SwapPrompt uses both image augmentation and target prompt with rich historical information to do adaption on the online prompt. Among them, image augmentation is a widely used design for constructing positive or negative pairs, in which the effectiveness has already been proved in many existing self-supervised contrastive learning frameworks. On the other hand, the same idea can be applied to textual modality by using EMA update strategies, which allows the target prompt to retain more historical information, and also be more stable and able to guide the tuning of the online prompt.
>
> Both methods are effective and indispensable, and the combination of the two methods naturally leads to "swap prediction", which we will further illustrate by analyzing why we do not use $L_1$ and $L_2$ as you mentioned:
>
> * If $L_1$ is used: the two prompts will adapt independently without any interaction so the online prompt cannot be updated under the guidance of target prompt. The advantage of historical information will be lost.
>
> * If $L_2$ is used: the contrastive phase is removed due to the lack of the approaching process between two augmented images. In this case, the advantage of image augmentation will diminish.
>
> We present the added experiment results of using $L_2$ or $L_2$ in the following table.
>
> |    | Caltech101  | DTD | Flowers102 | Oxford-Pets | UCF101 |
> |  :-  | :-:  | :-:  | :-:  | :-:  | :-:  |
> | $L_1$  | 87.38 | 47.22 | 69.63 | 87.71 | 64.68 |
> | $L_2$  | 88.45 | 46.69 | 69.28 | 87.30 | 64.46 |
> | SwapPrompt  | 89.90 | 47.34 | 70.22 | 89.14 | 65.66 |
>
> Note that our SwapPrompt uses swap prediction to take advantage of both image augmentation and historical information.
>
> **Q1:** "Could the authors ... Encoder Tuning?"
>
> **A1:** Prompt tuning generally works better than Image Encoder Tuning with small amounts of data. As the number of tuning parameters in prompt is far smaller than image encoder, prompt tuning requires less time compared to Image Encoder Tuning. On the other hand, directly altering a vision language model trained on extensive data could disturb its feature alignment between its image encoder and text encoder, possibly resulting in decreased performance. The results in Table 5 in [1] also show that fine-tuning the image encoder does not work well.
>
> [1] Zhou K, Yang J, Loy C C, et al. Learning to prompt for vision-language models[J]. International Journal of Computer Vision, 2022, 130(9): 2337-2348.
>
> **Q2:** "How about ... Image Encoder?"
>
> **A2:**  We added the experiment that SwapPrompt updates the parameters of all Batch Normalization (BN) layers in Image Enoder. The following table shows the results.
>
> |    | Caltech101  | DTD | Flowers102 | Oxford-Pets | UCF101 |
> |  :-  | :-:  | :-:  | :-:  | :-:  | :-:  |
> | SwapPrompt  | 89.90 | 47.34 | 70.22 | 89.14 | 65.66 |
> | update BN layers  | 86.65 | 38.18 | 54.35 | 76.83 | 58.29 |
>
> We also try to only update a part of norm layers but get worse results. Thus we think the adaptation on Image Enoder may lead to a decrease in performance.
>
> **Q3:** "Comparisons with ... also preferred."
>
> **A3:** Thanks for your valuable comments. As we explained in **A1&A2**, “only one online prompt” + “consistency maximization” forfeits the benefits of historical information. The experiments in **A1&A2** also demonstrated that using $l(p_1, p_2)$ does not yield better performance than SwapPrompt.
>
> **Q4:** "For Table 4... degrade a lot?"
>
> **A4:** The result of CLIP is the same as Table 1. When the batch size is 1, we will not do the data selection. The results of this experiment are presented in the following table. It can be seen that SwapPrompt still outperforms baselines.
>
> |  Batch size = 1  | Caltech101  | DTD | Flowers102 | Oxford-Pets | UCF101 |
> |  :-  | :-:  | :-:  | :-:  | :-:  | :-:  |
> | TPT  | 87.22 | 42.17 | 65.42 | 84.60 | 61.18 |
> |  UPL | 88.18 | 42.07 | 65.24 | 85.44 | 62.71 |
> | SwapPrompt  | 88.64 | 42.49 | 65.98 | 86.24 | 63.05 |
>
> **Q5:** SwapPrompt on corruption data (ImageNet-C)?
>
> **A5:** Thanks to the reviewer's suggestion, we added the experiments on ImageNet-C. Due to the limited time in rebuttal period, we cannot do experiments on all the corruption types, so we experimented on four types of corruption, which are **snow** in weather, **contrast** in digital,  **defocus blur** in blur and **gaussian noise** in noise, all of them have a corruption severity level of 3. The results of the experiments are shown in the following table. We can see that our method still has a good performance on the corruption types.
>
> |  **Method**   | Snow  | Contrast | Defocus blur | Gaussian noise |
> |  :-  | :-:  | :-: | :-:  | :-: |
> | CLIP  | 22.02 | 37.24 | 23.64 | 15.51 |
> | UPL | 25.85 | 41.11 | 25.09 | 17.01 |
> | SwapPrompt | 26.54 | 41.32 | 25.47 | 17.24 |
>
> **Q6:** "Could the authors ... an advantage."
>
> **A6:** Thanks for your valuable comments. We recorded the training time for SwapPrompt when its performance surpasses that of TPT (about 5 epochs' adaptation, refer to Figure 3 in the paper). The results show that SwapPrompt has lower GPU Memory utilization and shorter training time.
>
> | | Caltech101  | DTD | Flowers102 | Oxford-Pets | UCF101 |
> |  :-  | :-:  | :-: | :-:  | :-: | :-: |
> | TPT  | | | | |
> | Runtime (Min)  | 15.3 | 9.2 | 16.9 | 23.5 | 25.7 |
> | Memory (MiB) | 8985 | 9073 | 9023 | 9080 | 9074 |
> | SwapPrompt | | | | |
> | Runtime (Min)  | 4.3 | 4.1 | 4.2 | 3.9 | 4.2 |
> | Memory (MiB) | 4839 | 5042 | 4844 | 5092 | 4858 |
>
> **Q7:** Will the ... this work?
>
> **A7:** Yes, we will open the code once the paper is published.

---

> ### Author Response · Authors · 2023-08-14
> **Look Forward to Post-rebuttal**
>
> Dear Reviewer,
>
> Thank you very much for spending time providing constructive and valuable comments on our paper. We have carefully replied to your questions and presented the supplementary experiments in the response.
>
> The experimental results verify that the two variants of $L_{swap}$ ($L_1$ and $L_2$) cannot achieve the same performance as $L_{swap}$ and we explained the reasons. Besides, we discussed the adaptation of image encoder in test-time and provided some experimental results. We also provided more detailed studies to inspect the performance of SwapPrompt on **ImageNet-C** dataset, **small batch size**, and **efficiency** (i.e., GPU runtime and Memory).
>
> We would like to know if there are any additional clarifications or experiments that we can offer. We are looking forward to your feedback and sincerely invite you to update the score if concerns are addressed.
>
> Thanks a lot for your time again!
>
> Best, Authors

---

### Official Review · Reviewer_73P2 · 2023-07-07

**Soundness:** 3 good
**Presentation:** 3 good
**Contribution:** 3 good
**Rating:** 6
**Confidence:** 4

**Summary:**

The authors propose SwapPrompt, a novel test-time prompt adaptation method that leverages self-supervised contrastive learning to enhance prompt adaptation for pre-trained vision-language models. SwapPrompt employs an online prompt and an EMA updated target prompt, along with a swapped prediction mechanism, to facilitate prompt learning and improve performance. Experimental results demonstrate that SwapPrompt achieves state-of-the-art test-time adaptation performance on multiple datasets, and it can be easily deployed on vision-language models without additional requirements.

**Strengths:**

- The paper is well-written and presents information in a clear and coherent manner, making it easy to understand.
- The proposed method, SwapPrompt, can be easily implemented and used on test-time with vision-language models.
- The authors conducted extensive experiments on multiple datasets, and the results obtained were promising and showed positive outcomes.

**Weaknesses:**

There is no comparison for robustness to natural distribution shifts which is done in TPT.

**Questions:**

While the proposed method predominantly focuses on text prompts, there have been recent studies that utilize visual prompts or multimodal prompts (combining visual and text). I am curious to know what trends or patterns would emerge if visual prompts or multimodal prompts were employed in the proposed framework.

[1] Jia, M., Tang, L., Chen, B. C., Cardie, C., Belongie, S., Hariharan, B., & Lim, S. N. Visual prompt tuning. ECCV 2022
[2] Khattak, M. U., Rasheed, H., Maaz, M., Khan, S., & Khan, F. S. Maple: Multi-modal prompt learning. CVPR 2023

**Limitations:**

The authors did not explicitly address the limitations of the proposed method.

---

> ### Author Rebuttal · Authors · 2023-08-09
>
> Thanks for your valuable feedback and suggestions. We have tried to address all your concerns as follows:
>
> **W1:** There is no comparison for robustness to natural distribution shifts which is done in TPT.
>
> **A1:** If we understand your comment correctly, we have already shown the results of robustness to natural distribution shifts as you mentioned in TPT[1], i.e., Table 1. The experiment setting is essentially the same as Table 1 in TPT.
>
> Specifically, natural distribution shifts in TPT refer to training with ImageNet and testing with four different ImageNet variants. While TPT does not require training data, its prompt is initialized as "a photo of a". Similar to TPT, our method does not require training data as well, thus we also did not use prompts trained on ImageNet. The comparisons in our Table 1 are fair. The following excerpt from the results in table 1 shows the results.
>
> |  **Method**   | ImageNet  | ImageNet-V2 | ImageNet-A | ImageNet-R | ImageNet-Sketch |
> |  :-  | :-: | :-:  | :-:  | :-: | :-: |
> | CLIP  | 58.18 | 51.36 | 21.69 | 55.98 | 33.33 |
> | UPL | 61.19 | 52.07 | 23.59 | 57.09 | 36.40 |
> | TPT | 60.74 | 54.35 | 26.24 | 58.72 | 35.02 |
> | SwapPrompt | 61.80 | 53.94 | 24.46 | 60.88 | 38.21 |
>
> [1] Shu M, Nie W, Huang D A, et al. Test-time prompt tuning for zero-shot generalization in vision-language models[J]. Advances in Neural Information Processing Systems, 2022, 35: 14274-14289.
>
> **Q2:** While the proposed method predominantly focuses on text prompts, there have been recent studies that utilize visual prompts or multimodal prompts (combining visual and text). I am curious to know what trends or patterns would emerge if visual prompts or multimodal prompts were employed in the proposed framework.
>
> **A2:** Thanks for your valuable comments. We believe that our proposed SwapPrompt will be also effective on visual prompts or multimodal prompts.  The principle behind our method is to utilize the powerful potentials of self-supervised learning in fine-tuning the prompt on test time, which can be considered as a general framework for prompt-based tuning methods. In future work, we would like to consider changing our method directly to do adaption on visual prompts.

---

> ### Author Response · Authors · 2023-08-14
> **Look Forward to Post-rebuttal**
>
> Dear reviewer,
>
> Thanks for your valuable comments on our paper. We have tried our best to answer your concerns. If you have any new questions, please let me know. So we can address them as soon as possible within the rolling discussion.
>
> Thanks a lot for your time again!
>
> Best, Authors

---

> > ### Comment · Reviewer_73P2 · 2023-08-21
> > **Response to Rebuttal**
> >
> > After reading all reviews and the authors' rebuttal I personally still lean towards acceptance of the paper.

---

### Official Review · Reviewer_VTrP · 2023-07-07

**Soundness:** 3 good
**Presentation:** 3 good
**Contribution:** 2 fair
**Rating:** 5
**Confidence:** 5

**Summary:**

This paper proposes the test-time prompt adaption method "SwapPrompt" for the vision-language model. The author leverages contrastive learning strategy to learn the prompts on multiple augmented views of unseen data. To better adapt to out-of-distribution tasks, the learnable target and online prompts are applied to samples and interact with each other to retrieve the swapped prediction loss. Besides, a pseudo-labeling strategy and data selection process is applied to the test-time data. The results show that SwapPrompt has a better performance compared to other unsupervised prompt learning baselines.

**Strengths:**

The paper is well-organized and easy to follow. The reviewer thought the idea of using swapped prediction loss to optimize the prompt was good.

**Weaknesses:**

The related works might lack some existing works. For example, one of the similar works that the author didn't mention [1] also applied the multiple augmented views of samples for calculating the marginal cross-entropy loss to do the adaptation.

The benefit of applying the EMA update strategy on the prompt is not clear enough. The author mentions that the target prompts are slowly updated by the online prompt for the purpose of learning more knowledge from prior pre-trained models, but it seems like target prompts are updated after every training step (equation 3). Does one training step here mean a one-time adaption for just one batch or multiple-round adaption on multiple batches?

For the experiment part, why not compare with [1], which also uses multiple augmented views of samples and unsupervised objective function to improve test-time adaptation?  It seems like the idea is similar to this work which adapts the model by minimizing the marginal output distribution across the augmentations.

[1] MEMO: Test Time Robustness via Adaptation and Augmentation. Marvin Zhang, Sergey Levine, Chelsea Finn

**Questions:**

To my understanding, the proposed SwapPrompt can better leverage the prior knowledge of the pre-trained model during the adaption by using the swap prediction mechanism and pseudo labeling strategy. However, the experiment results might not convince the reviewer that SwapPrompt is better than the current method. Also, there exist some other kinds of OOD benchmarks, such as the ImageNet-C. Can SwapPrompt also be leveraged on multiple kinds of corruption types?

**Limitations:**

See abovementioned weakness and questions.

---

> ### Author Rebuttal · Authors · 2023-08-09
>
> Thanks for your valuable feedback and suggestions. We have tried to address all your comments and suggestions on Weaknesses and Questions as follows:
>
> **W1:** The related works might lack some existing works. For example, one of the similar works that the author didn't mention [1] also applied the multiple augmented views of samples for calculating the marginal cross-entropy loss to do the adaptation.
>
> **A1:** We agree with the reviewer that both MEMO in [1] and our SwapPrompt utilize multiple augmented views of samples. However, MEMO relies on the **marginal entropy minimization** rather than the **marginal cross-entropy loss** employed in our work. This makes MEMO more similar to TPT in the baseline, because both MEMO and TPT use multiple augmented views of samples and entropy minimization.
>
> In fact, by comparing the published source code of MEMO and TPT, we found that TPT uses almost exactly the same loss function as MEMO, the difference between these two approaches is that MEMO adapts all parameters of a network model, while TPT only tunes the prompt.
>
> We added experiments to combine CLIP with MEMO: we used MEMO's training method and the image encoder in CLIP, and fine tuned the image encoder at test time. The detailed results can be shown in the following table. From the experimental results, we can observe that SwapPrompt also outperforms MEMO.
>
> |  **Method**   | Caltech101  | DTD | Flowers102 | Oxford-Pets | UCF101 |
> |  :-  | :-:   | :-:  | :-:   | :-:   | :-:   |
> | CLIP  | 85.13 | 42.16 | 65.40 | 83.05 | 61.15 |
> | **MEMO**  | 85.44 | 42.47 | 62.63 | 83.62 | 61.28 |
> | UPL | 86.37 | 45.04 | 67.11 | 88.53 | 63.63 |
> | TPT | 87.22 | 42.17 | 65.42 | 84.60 | 61.18 |
> | SwapPrompt | 89.90 | 47.34 | 70.22 | 89.14 | 65.66 |
>
> Besides that, we improved the presentation of “Related Work” by conducting a more comprehensive literature review in the new version.
>
> **W2:** The benefit of applying the EMA update strategy on the prompt is not clear enough. The author mentions that the target prompts are slowly updated by the online prompt for the purpose of learning more knowledge from prior pre-trained models, but it seems like target prompts are updated after every training step (equation 3). Does one training step here mean a one-time adaption for just one batch or multiple-round adaption on multiple batches?
>
> **A2:** Applying the EMA update strategy on the prompt is to allow the target prompt to retain more historical information, and be more stable and able to guide the tuning of online prompt. Table 3 in the paper shows the experiment results without using the EMA update strategy, and it shows that EMA is an indispensable part to achieve better performance.
>
> “Target prompts are updated after every training step (equation 3)." The training step here refers to a one-time adaptation for just one batch. The purpose is to make target prompts update more slowly (compared to online prompts) at an appropriate rate, so we set a fairly large delay rate of target prompts to ensure that the target prompt retains a large amount of history information.  At the same time, the decay rate can't be 1, because then the target prompt doesn't update, and it cannot learn from the test data.
>
> Regarding your comment about updating the target prompt after multiple-round adaptation on multiple batches, this is an interesting idea, and we believe that this approach also lowers the update rate of the target prompt, which we would like to discuss further in the future work.
>
> **W3:** For the experiment part, why not compare with [1], which also uses multiple augmented views of samples and unsupervised objective function to improve test-time adaptation? It seems like the idea is similar to this work which adapts the model by minimizing the marginal output distribution across the augmentations.
>
> **A3:** As answered in **A1**, MEMO is a baseline worth to compare, we thank the reviewer for his constructive comments and the experimental results can be found in **A1**.
>
> **Q4:** To my understanding, the proposed SwapPrompt can better leverage the prior knowledge of the pre-trained model during the adaption by using the swap prediction mechanism and pseudo labeling strategy. However, the experiment results might not convince the reviewer that SwapPrompt is better than the current method. Also, there exist some other kinds of OOD benchmarks, such as the ImageNet-C. Can SwapPrompt also be leveraged on multiple kinds of corruption types?
>
> **A4:** Thanks to the reviewer's suggestion, we added the experiments on **ImageNet-C**. Due to limited time in rebuttal period, we cannot do experiments on all the corruption types, so we experimented on four types of corruption, which are **snow** in weather, **contrast** in digital,  **defocus blur** in blur and **gaussian noise** in noise, all of them have a corruption severity level of 3. The results of the experiments are shown in the following table. We can see that our method still has a good performance on the corruption types.
>
> |  **Method**   | Snow  | Contrast | Defocus blur | Gaussian noise |
> |  :- | :-:   | :-:   | :-:   | :-:   |
> | CLIP  | 22.02 | 37.24 | 23.64 | 15.51 |
> | UPL | 25.85 | 41.11 | 25.09 | 17.01 |
> | TPT | 26.33 | 40.64 | 25.12 | 16.89 |
> | SwapPrompt | 26.54 | 41.32 | 25.47 | 17.24 |

---

> ### Author Response · Authors · 2023-08-14
> **Look Forward to Post-rebuttal**
>
> Dear Reviewer,
>
> Thank you very much for spending time providing constructive and valuable comments on our paper. We have carefully replied to your questions and presented the supplementary experiments in the response.
>
> The experimental results verify that the proposed SwapPrompt can outperform the existing methods (i.e., **MEMO**) in test time adaptation of vision-language model. Besides, we provided the results on **ImageNet-C** dataset. We also discussed the EMA update strategy and explained why the target prompt can be updated after every training step.
>
> We would like to know if there are any additional clarifications or experiments that we can offer. We are looking forward to your feedback and sincerely invite you to update the score if concerns are addressed.
>
> Thanks a lot for your time again!
>
> Best, Authors

---

> > ### Comment · Reviewer_VTrP · 2023-08-18
> >
> > Thanks for the authors' response and effort. The authors' response has addressed most of my concerns. The results show the proposed method can outperform other baselines, such as MEMO and can be applied to other  OOD benchmarks, which are convincing. Therefore, I decided to increase my rating to borderline accept.

---

> > > ### Author Response · Authors · 2023-08-19
> > >
> > > Dear reviewer,
> > >
> > > Thanks for your valuable comments on our paper. We will conduct more comprehensive explanations and experiments for your concerns in the upcoming version.
> > >
> > > Thanks a lot for your time again!
> > >
> > > Best, Authors

---

### Author Response · Authors · 2023-08-19
**Author General Response**

We sincerely appreciate all reviewers’ efforts in reviewing our paper and for the constructive feedback. Here, apart from the response to each reviewer, we would like to thank reviewers for the acknowledgment of our work and highlight new results added during the rebuttal.

We are encouraged that the reviewers appreciate and recognize our contributions:
* The problem of test-time prompt adaptation on pre-trained vision-language backbones is quite a major and interesting challenge. [V8X4]
* The proposed SwapPrompt method is novel in the TTA community. It successfully introduces self-supervised contrastive learning into Prompt Tuning-based methods. [VTrP, N4YN, V8X4]
* Detailed experiments on ImageNet, its four variants and nine other benchmark datasets verify the superiority of our proposed SwapPrompt over existing methods. [73P2, N4YN, V8X4, 3xii]
* The proposed method can be easily implemented and used during test time. [73P2]
* The paper is technically sound and easy to understand. [VTrP, 73P2, N4YN, V8X4, 3xii]

In this rebuttal, we have added more explanations and supporting results following the reviewers’ suggestions.
* We discuss the benefit of EMA update strategy and explained why the target prompt can be updated after every training step. [VTrP]
* We add more comparison experiments to verify the effectiveness of SwapPrompt from two perspectives: 1) **Imagenet-C** and **FGVCAircraft** datasets; 2) another test time adaptation method: **MEMO**. [VTrP, N4YN, 3xii]
* We conduct more detailed ablation studies on **small batch size**, **hyperparameters** **α** and **β**, and different **augmentation** **methods** and only loss function **$L_{swap}$** in Table 2. We also compare the **efficiency** between SwapPrompt and TPT. [N4YN, 3xii]
* We explain the necessity of the swap mechanism. Compared with other variants of $L_{swap}$ , $L_{swap}$ can take advantage of both image augmentation and historical information. [N4YN]

Thank you for your time again !

---

### Decision · Program_Chairs · 2023-09-21

**Decision:**

Accept (poster)

**Comment:**

This work argues that existing test-time adaptation (TTA) methods cannot fully exploit the representation capabilities of pre-trained models. To this end, the authors proposed a new method that leverages the self-supervised contrastive learning to facilitate the test-time prompt adaptation.

Reviewers recognize the importance of this problem, novelty of the proposed method and the effectiveness of the methods.

In the rebuttal the author has addressed the concerns regarding the requirements of more detailed comparisons and ablations, along with other technical details. Based on the rebuttal, two reviewers raised their scores. The final scores are unanimous accept, albeit with different degrees (5, 5, 5, 6, 8).

Overall, we feel that that the major concerns of the reviewers were addressed by the author. The proposed method is interesting and effective. We therefore recommend accept. In the final version, please add the additional experiments and discussions.